# Evaluation of symptomatology and viral load among residents and healthcare staff in long-term care facilities: A coronavirus disease 2019 retrospective case-cohort study

**Mitch van Hensbergen** [1,2]*, **Casper D. J. den Heijer**[1,2,3], **Suhreta Mujakovic**[1], **Nicole H. T. M. Dukers-Muijrers**[1,4], **Petra F. G. Wolffs**[3], **Inge H. M. van Loo**[3], **Christian J. P. A. Hoebe**[1,2,3]

1 Department of Sexual Health, Infectious Diseases, and Environmental Health, South Limburg Public Health Service, Limburg, The Netherlands, 2 Faculty of Health, Medicine and Life Sciences, Department of Social Medicine, Care and Public Health Research Institute (CAPHRI), Maastricht University, Maastricht, Limburg, The Netherlands, 3 Faculty of Health, Medicine and Life Sciences, Department of Medical Microbiology, Care and Public Health Research Institute (CAPHRI), Maastricht University Medical Centre (MUMC+), Maastricht, Limburg, the Netherlands, 4 Faculty of Health, Medicine and Life Sciences, Department of Health Promotion, Care and Public Health Research Institute (CAPHRI), Maastricht University, Maastricht, Limburg, the Netherlands

* Mitch.vanhensbergen@ggdzl.nl

## Abstract

### Objectives

We evaluated COVID-19 symptoms, case fatality rate (CFR), and viral load among all Long-Term Care Facility (LTCF) residents and staff in South Limburg, the Netherlands (February 2020-June 2020, wildtype SARS-CoV-2 Wuhan strain).

### Methods

Patient information was gathered via regular channels used to notify the public health services. Ct-values were obtained from the Maastricht University Medical Centre laboratory. Logistic regression analyses were performed to assess associations between COVID-19, symptoms, CFR, and viral load.

### Results

Of 1,457 staff and 1,540 residents, 35.1% and 45.2% tested positive for COVID-19. Symptoms associated with COVID-19 for female staff were fever, cough, muscle ache and loss of taste and smell. Associated symptoms for men were cough, and loss of taste and smell. Associated symptoms for residents were subfebrility, fatigue, and fever for male residents only. LTCF residents had a higher mean viral load compared to staff. Male residents had a higher CFR (35.8%) compared to women (22.5%). Female residents with Ct-values 31 or less had increased odds of mortality.

**Data Availability Statement:** The generated dataset analysed during the current study is

available for registered users of Synapse in the Synapse repository at https://www.synapse.org/#!Synapse:syn26939703/, under the Synapse ID 'syn26939703' and DOI: 10.7303/syn26939703.1.

**Funding:** The author(s) received no specific funding for this work.

**Competing interests:** The authors have declared that no competing interests exist.

## Conclusions

Subfebrility and fatigue seem to be associated with COVID-19 in LTCF residents. Therefore, physicians should also consider testing residents who (only) show aspecific symptoms whenever available resources prohibit testing of all residents. Viral load was higher in residents compared to staff, and higher in male residents compared to female residents. All COVID-19 positive male residents, as well as female residents with a medium to high viral load (Ct-values 31 or lower) should be monitored closely, as these groups have an overall increased risk of mortality.

## Introduction

COVID-19 outbreaks have shown to cause a high burden of disease and deaths within long-term care facilities for the elderly (LTCFs) [1–5]. At the time of the first COVID-19 wave in the Netherlands (early March 2020, wildtype SARS-CoV-2 Wuhan strain), some European countries reported over 60% of all COVID-19 related deaths to occur within LTCFs [6]. Although this number has decreased since vaccination has been introduced within LTCFs [7], the proportion of COVID-19-related deaths still ranges between 0% and 62% in LTCF residents in European countries, as reported by the European Centre for Disease Prevention and Control until November 9[th] 2021 [8]. LTCFs are typified by a frail and vulnerable population with a high care demand [9,10], in which residents often have chronic diseases, mental impairment, and complex health needs [6,11–13]. Moreover, it is challenging to recognize COVID-19 in an early phase due to a wide range of aspecific symptoms [4,6,12–21], as well as asymptomatic cases [1,3,4,16–19,22,23]. Hence, transmission often takes place before adequate control measures have been taken. Although a multitude of studies have compared the prevalence of symptoms between COVID-19 negative and COVID-19 positive LTCF residents, most had a limited number of residents or observed symptoms [1,24–29]. Additionally, few studies have examined the relationship between viral load and symptoms.

Viral load can be used as an indication for the degree of infectiousness and the prognosis of COVID-19 [26,30–32]. LTCFs would benefit from a better understanding of which symptoms are related to different levels of viral load, as well as how viral load differs between LTCF staff and residents, for men and women. The cycle threshold value (Ct-value) of a Real-Time Polymerase Chain Reaction (RT-PCR) test can be used as an indicator to estimate the viral load. Understanding the symptomatology of LTCF staff and residents in relation to viral load could further specify guidelines for prevention and control of COVID-19 in LTCFs.

The objectives of our study were to assess symptoms associated with COVID-19 in LTCF staff and residents and to examine the relationship between viral load and COVID-19-related symptoms and case fatality in LTCF residents and staff in the South Limburg region.

## Materials and methods

### Study design

We performed an epidemiological and laboratory analysis of LTCF residents and staff who were tested for SARS-CoV-2 in South Limburg, the Netherlands, from February 27[th] 2020 to June 1[st] 2020.

## Case definition and testing policy

When the first COVID-19 patient was confirmed in the Netherlands on the 27[th] of February 2020, the suspected case definition included a sudden onset of fever (38 degrees Celsius), paired with cough or shortness of breath. In addition to these symptoms, a suspected individual must have had a history of travel or residence in a country/area reporting local or community transmission (defined through a large number of cases which cannot be linked to transmission chains or multiple unrelated clusters in several areas of the country/area), or have had to be in close contact with a confirmed, or probable COVID-19 case in the past two weeks. Contacts of a confirmed COVID-19 case were tested only when they were part of a vulnerable population group; people who had a higher risk of severe COVID-19 outcomes, such as people aged 70 and older, as well as people with underlying disease [33,34]. Within Dutch hospitals, patients were also suspected of COVID-19 when they were diagnosed with pneumonia with unknown cause irrespective of an epidemiological link [5].

After widespread circulation of SARS-CoV-2 was observed in the Netherlands (halfway March 2020), the travel link as requirement for the suspected case definition was omitted. Due to the scarce number of tests available, testing outside the hospital was reserved exclusively for vulnerable individuals when their physician deemed a test to be necessary for further treatment. In the LTCF setting, the main reason for testing was to determine whether SARS-CoV-2 introduction had taken place. Whenever two residents tested positive within the same ward, further testing was deemed unnecessary and it was assumed that a COVID-19 outbreak was ongoing at that ward. From April 6[th] 2020, nationwide testing was expanded to also include vulnerable groups. These groups were tested whenever they displayed symptoms of disease, in order to safely receive care or treatment. Tests were only available for healthcare staff who were essential in providing care and when they showed symptoms indicative of COVID-19, which at the time included symptoms of a cold, such as a runny nose, sneezing, sore throat, cough, shortness of breath, subfebrility or fever, and a sudden loss of smell and/or taste [33]. During our study period, testing was only performed in individuals with symptoms, meaning no routine testing or testing for asymptomatic cases was done.

## Information and sample gathering

Prior to the test, information about the individual was gathered by phone, including date of birth, sex, symptoms, comorbidities, date of onset, profession, and whether they were a LTCF resident or LTCF staff member. Until April 17[th], all regionwide COVID-19 tests outside of the hospital were performed by our PHS. However, as of April 17[th] 2020, LTCFs had the option of performing COVID-19 tests themselves as more tests became available; nasopharyngeal and throat swabs from COVID-19 suspected LTCF residents were collected by LTCF staff and sent to the microbiological lab for PCR-testing. However, the test results and patient information would still be communicated to the PHS via an online registration form. Negatively tested LTCF residents were also reported to the PHS, but no additional information on comorbidities or symptoms were provided. Finally, all deaths among residents with a strong suspicion of COVID-19 were reported to the PHS in the event that no positive COVID-19 test was available at the time of death.

## Study population and selection

By combining all reports of staff and residents suspected of COVID-19, a preselection of 3804 cases was made. If a person was tested more than once and results differed between tests, the registration of the (first) positive test was kept, and the registration for the negative test(s) and subsequent positive test(s) were discarded (within a period of 8 weeks). If a person only had

negative tests, only the registration of the first negative test was included. Additional symptoms of disease which appeared in between positive tests were added to the initial registration in order to obtain the full spectrum of symptoms.

## Laboratory analysis

Nasopharyngeal and throat swabs were taken from residents and staff suspected of COVID-19. RT-PCR was used for the detection of SARS-CoV-2 [35]. RNA was extracted from the samples by automated total nucleic acid extraction using the MP96 (Roche Diagnostics, Rotkreuz, Switzerland) per the manufacturer's instructions. In-house RT-PCR was performed using Quantstudio 5 (Applied Biosystems, MA, USA), based on a PCR published by Corman et al. [35] targeting the E-gene. For PCR, a 20 microliter PCR reaction was used, including 5 microliter Taqpath 1-step RT mastermix (Applied Biosystems), 100–800 nM of primers and probes and 10 microliter extracted RNA. All samples were spiked with murine cytomegalovirus RNA before extraction, which was used as an extraction and amplification control. For all tests done by MUMC+ the Ct-values were also determined and registered.

## Statistical analysis

Baseline characteristics were compared between those with positive and negative tests separately for residents and staff, by using two-sided independent samples t-tests for continuous variables and chi-square tests for categorical variables. Because several calls urged for sex-specific information in COVID-19 studies, as well as an increased risk of mortality for male patients with COVID-19, analyses were stratified by sex [36,37]. A Mann-Whitney U-test was performed to evaluate the difference in Ct-value between staff and residents, and men and women. Furthermore, to visualize the number of tests (negative and positive) for the study period, we constructed an epidemiological curve (epicurve) for all COVID-19 cases from February 27th up to and including June 1st. If the date of onset was not available, the date which resembled the date of onset closest was chosen. We prioritized the dates as followed: day of onset, day of testing, day of test result, day of communication to the PHS.

In the evaluation of which symptoms were associated with a positive test result, the following symptoms were included: fever (38.0 Celsius and above) and subfebrility (37.5 to 38 degrees Celsius), cough, sore throat, runny nose, dyspnoea, fatigue, muscle ache, headache, diarrhoea, loss of smell or taste, malaise, and nausea. Cases for which no registry was made (i.e. lacking a 'yes' or 'no' for a symptom) were recoded to a 'no'. This was done for 959 cases (30%). The remaining 70% of cases did not have any empty registries and therefore did not require recoding.

Each symptom was analysed using univariate logistic regression models. Variables with statistically significant ($\alpha \leq 0.05$) odds ratios (ORs) were entered into the multivariable logistic regression model using the forced entry method, which also included age as a covariate, resulting in adjusted ORs (aOR) for each symptom. Included variables were assessed for colinearity and interaction. Correlations between variables included in the multivariable model were <0.14, and VIFs were all <1.1. Additionally, we calculated the CFR including, as well as excluding, the highly suspected COVID-19 deaths. Whenever it was not possible to calculate the OR because no case with that symptom was present in either the COVID-19 negative or the COVID-19 positive group, a temporary case we imputed with average values for the other variables.

Finally, multinominal multivariable regression analyses were performed to determine the association between symptoms and Ct-value level, corrected for age. We analysed each statistically significant symptom from the multivariable regression analysis with Ct-value categorized

into tertiles. The cut-off values for these tertiles were based on the overall range of Ct-values found for residents and staff. The Ct-values were categorized into: 'low viral load' (Ct>31), 'medium viral load' (22< Ct≤31), and 'high viral load' (Ct≤22). This was done separately for staff and residents. In addition, we determined the CFR for each Ct-value tertile. All analyses were stratified for sex and conducted using IBM SPSS Statistics version 26 (IBM, Armonk, NY, USA).

### Ethics approval and consent to participate

The Medical Ethics Committee of Maastricht University Medical Centre (MUMC+) exempted this study from official approval under prevailing laws in the Netherlands after official review (METC number: 2021–2901).

## Results

### Descriptive data

After merging the data, 3303 unique residents and staff remained. A total of 139 residents and staff were wrongly categorized as belonging to LTCFs and were excluded from the dataset, resulting in a dataset consisting of 3164 positively and negatively tested staff (n = 1461) and residents (n = 1703). Test results were known for 1,457 (99.7%) staff and 1,540 (90.4%) residents. Furthermore, 11 cases were tested despite a lack of symptoms, which goes against testing policy at the time. Although these cases were included in the descriptive analysis, these cases were excluded from the logistical analyses, along with 779 cases from whom no symptom data was available (See flow chart in Fig 1).

The mean age of staff was 42.0 years (range: 16 to 68 years); for residents the mean age was 83.9 years (range:21 to 104 years). The most reported comorbidities in COVID-19 positive residents were cardiovascular disease, dementia/Alzheimer, and diabetes. These characteristics of

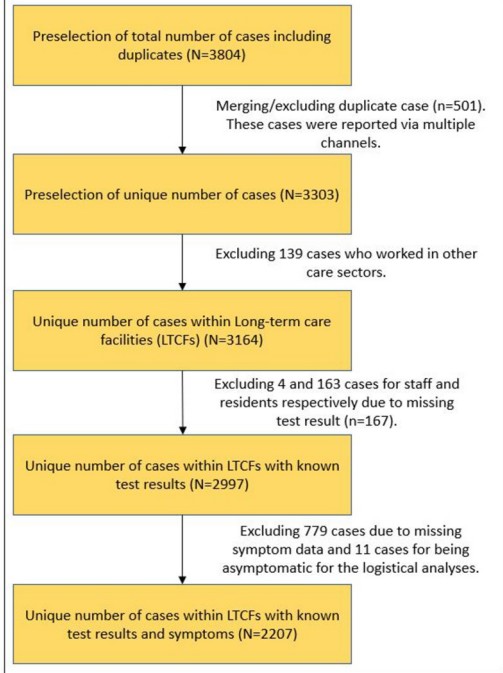

**Fig 1. Flowchart of the selection process for the dataset.**

our study population are shown in the supplemental S1 Table. Descriptive statistics on cases without a test result are shown in the supplemental S2 Table. Due to the high number of missing data for comorbidities, these data were not included in further analyses.

For residents, the overall CFR for confirmed COVID-19 cases was 26.6% (185/696), of which men had a statistically significant higher CFR of 35.8% (77/215) compared to women, who had a CFR of 22.5% (108/480). When we included the highly suspected COVID-19 deaths, the overall CFR increased to 40.4% (347/859), of which men had a statistically significant higher CFR of 53.6% (159/297) compared to women, who had a CFR of 33.5% (188/561). There were no recorded deaths among LTCF staff.

## The epidemiological curve (epicurve)

The epicurve for all confirmed COVID-19 cases in this study's time period is shown in Fig 2, including all relevant timepoints concerning the testing policy. Initially, most cases were seen among residents. Later on, cases were more equally distributed between staff and residents.

## Multivariable logistic regression analysis

The associations of symptoms with COVID-19 are displayed in Table 1A for staff and in Table 1B for residents. Symptoms associated with COVID-19 for female staff were fever (aOR 2.13 CI 1.66–2.75), cough (aOR 1.51 CI 1.14–1.99), muscle ache (aOR 2.75 CI 1.86–4.08) and loss of taste and smell (aOR 5.57 CI 3.80–8.17), whereas associated symptoms for men were cough (aOR 2.55 CI 1.12–5.82) and loss of taste and smell (aOR 3.98 CI 1.40–11.36) only. For residents these were subfebrility (aOR 11.96 CI 1.47–97.12 for men, aOR 5.22 CI 1.93–14.12 for women), fatigue (aOR 10.60 CI 1.37–81.72 for men, aOR 6.04 CI 2.13–17.14 for women), and fever (aOR 2.33 CI 1.17–4.63) for male residents only.

## Ct-values

The Ct-values for positive staff and residents are shown in Fig 3. Ct-values were unknown for 190 (37.2%) and 76 (11.1%) positively tested staff and residents, respectively. The mean Ct-value was higher for LTCF staff (28.3, standard deviation (SD): 6.1)) compared to LTCF residents (26.4, SD: 6.6, α ≤ 0.001). On average, male residents reported a lower Ct-value (25.2,

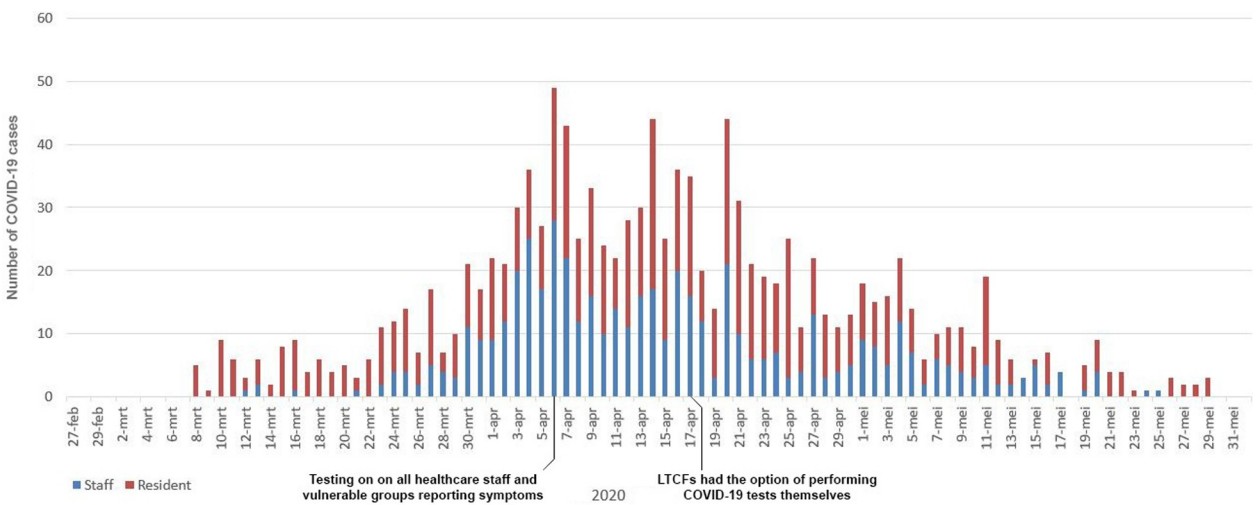

**Fig 2. Epidemiological curve of COVID-19 cases by generated starting date, the Netherlands, 27 February –1 June 2020 (n = 1,208).**

**Table 1.** a: Associations in odds ratios (OR) between reported symptoms and COVID-19 positivity for LTCF staff, corrected for age and stratified by sex. b: Associations in odds ratios (OR) between reported symptoms and COVID-19 positivity for LTCF residents, corrected for age and stratified by sex.

**A**

| | LTCF staff with known symptoms and test results (n = 1,404) | | | | | | | | | | | |
| | COVID-19- (n = 897) | | | | COVID-19+ (n = 503) | | | | | | | |
| | Male (n = 90) | | Female (n = 807) | | Male (n = 59) | | Female (n = 444) | | Male | | Female | |
| | n/N | % | n/N | % | n/N | % | n/N | % | Unadjusted OR (95% CI) | Adjusted OR (95% CI) | Unadjusted OR (95% CI) | Adjusted OR (95% CI) |
| Symptoms | | | | | | | | | | | | |
| Temp. Increase | 41/90 | 45.5 | 321/807 | 39.8 | 33/59 | 55.9 | 246/444 | 55.4 | | | | |
| Subfebrility[a] | 1/90 | 1.1 | 21/807 | 2.6 | 0/59 | 0.0 | 7/444 | 1.6 | 1.89 (0.11–31.38) | 1.45 (0.08–26.37) | 0.82 (0.34–1.96) | 0.93 (0.38–2.27) |
| Fever | 40/90 | 44.4 | 300/807 | 37.2 | 33/59 | 55.9 | 239/444 | 53.8 | 1.56 (0.80–3.02) | 1.74 (0.85–3.56) | **1.96 (1.54–2.48)** | **2.13 (1.66–2.75)** |
| Cough | 58/90 | 64.4 | 545/807 | 67.5 | 46/59 | 78.0 | 330/444 | 74.3 | 1.95 (0.92–4.14) | **2.55 (1.12–5.82)** | **1.39 (1.07–1.80)** | **1.51 (1.14–1.99)** |
| Sore throat | 42/90 | 46.7 | 426/807 | 52.8 | 19/59 | 32.2 | 211/444 | 47.5 | 0.54 (0.27–1.08) | - | 0.81 (0.64–1.02) | - |
| Runny nose | 40/90 | 44.4 | 386/807 | 47.8 | 31/59 | 52.5 | 220/444 | 49.5 | 1.38 (0.72–2.67) | - | 1.07 (0.85–1.35) | - |
| Dyspnoea | 36/90 | 40.0 | 319/807 | 39.5 | 24/59 | 40.7 | 165/444 | 37.2 | 1.03 (0.53–2.01) | - | 0.91 (0.71–1.15) | - |
| Fatigue | 16/90 | 17.8 | 146/807 | 18.1 | 12/59 | 20.3 | 86/444 | 19.4 | 1.18 (0.51–2.72) | - | 1.09 (0.81–1.46) | - |
| Muscle ache | 9/90 | 10.0 | 54/807 | 6.7 | 10/59 | 16.9 | 75/444 | 16.9 | 1.84 (0.70–4.84) | 1.77 (0.63–5.82) | **2.83 (1.96–4.11)** | **2.75 (1.86–4.08)** |
| Headache | 25/90 | 27.8 | 241/807 | 29.9 | 14/59 | 23.7 | 152/444 | 34.2 | 0.81 (0.38–1.72) | - | 1.22 (0.96–1.57) | - |
| Diarrhoea[a] | 6/90 | 6.7 | 36/807 | 4.5 | 0/59 | 0.0 | 20/444 | 4.5 | 0.24 (0.03–2.02) | - | 1.01 (0.58–1.77) | - |
| Loss of smell/taste | 8/90 | 8.9 | 51/807 | 6.3 | 12/59 | 20.3 | 101/444 | 22.7 | **2.62 (1.00–6.86)** | **3.98 (1.40–11.36)** | **4.37 (3.05–6.26)** | **5.57 (3.80–8.17)** |
| Malaise | 1/89 | 1.1 | 9/807 | 1.1 | 1/59 | 1.7 | 8/444 | 1.8 | 1.53 (0.09–25.02) | - | 1.63 (0.62–4.25) | - |
| Nausea | 4/90 | 4.4 | 34/807 | 4.2 | 2/59 | 3.4 | 14/444 | 3.2 | 0.75 (0.13–4.26) | - | 0.74 (0.39–1.40) | - |

**B**

| | LTCF residents with known symptoms and test results (n = 803) | | | | | | | | | | | |
| | COVID-19- (n = 218) | | | | COVID-19+ (n = 548) | | | | | | | |
| | Male (n = 55) | | Female (n = 163) | | Male (n = 178) | | Female (n = 370) | | Male | | Female | |
| | n/N | % | n/N | % | n/N | % | n/N | % | Unadjusted OR (95% CI) | Adjusted OR (95% CI) | Unadjusted OR (95% CI) | Adjusted OR (95% CI) |
| Symptoms | | | | | | | | | | | | |
| Temp. Increase | 34/55 | 61.8 | 119/163 | 73.0 | 139/178 | 78.1 | 279/370 | 75.4 | | | | |
| Subfebrility[a] | 0/55 | 0.0 | 5/163 | 3.1 | 19/178 | 10.7 | 54/370 | 14.6 | **10.23 (1.28–81.87)** | **11.96 (1.47–97.12)** | **5.22 (1.95–13.97)** | **5.22 (1.93–14.12)** |
| Fever | 34/55 | 61.8 | 114/163 | 69.9 | 120/178 | 67.4 | 225/370 | 60.8 | 1.90 (0.99–3.65) | **2.33 (1.17–4.63)** | 0.95 (0.62–1.46) | 1.04 (0.67–1.61) |
| Cough | 37/55 | 67.3 | 111/163 | 68.1 | 108/178 | 60.7 | 216/370 | 58.4 | 0.75 (0.40–1.42) | 0.68 (0.35–1.33) | **0.66 (0.45–0.97)** | 0.68 (0.46–1.02) |
| Sore throat | 2/55 | 3.6 | 11/163 | 6.7 | 10/178 | 5.6 | 23/370 | 6.2 | 1.58 (0.34–7.43) | - | 0.92 (0.44–1.93) | - |
| Runny nose | 6/55 | 10.9 | 23/163 | 14.1 | 26/178 | 14.6 | 51/370 | 13.8 | 1.40 (0.54–3.59) | - | 0.97 (0.57–1.66) | - |
| Dyspnoea | 22/55 | 40.0 | 64/163 | 39.3 | 73/178 | 41.0 | 139/370 | 37.6 | 1.04 (0.56–1.93) | - | 0.93 (0.64–1.36) | - |
| Fatigue | 1/55 | 1.8 | 4/163 | 2.5 | 25/178 | 14.0 | 51/370 | 13.8 | **8.82 (1.17–66.70)** | **10.60 (1.37–81.72)** | **6.36 (2.26–17.90)** | **6.04 (2.13–17.14)** |
| Muscle ache[a] | 0/55 | 0.0 | 1/163 | 0.6 | 1/178 | 0.6 | 12/370 | 3.2 | 0.31 (0.02–5.05) | - | 5.43 (0.70–42.11) | - |
| Headache[a] | 0/55 | 0.0 | 0/163 | 0.0 | 4/178 | 2.2 | 16/370 | 4.3 | 1.26 (0.14–11.55) | - | 7.37 (0.97–56.03) | - |
| Diarrhoea[a] | 0/55 | 0.0 | 6/163 | 3.7 | 10/178 | 5.6 | 31/370 | 8.4 | 3.27 (0.41–26.15) | - | 2.39 (0.98–5.85) | - |
| Loss of smell/taste[a] | 0/55 | 0.0 | 3/163 | 1.8 | 4/178 | 2.2 | 9/370 | 2.4 | 1.26 (0.14–11.55) | - | 1.33 (0.36–4.98) | - |
| Malaise | 5/55 | 9.1 | 14/163 | 8.6 | 15/178 | 8.4 | 34/370 | 9.2 | 0.92 (0.32–2.66) | - | 1.08 (0.56–2.07) | - |
| Nausea[a] | 0/55 | 0.0 | 3/163 | 1.8 | 6/178 | 3.4 | 6/370 | 1.6 | 1.92 (0.23–16.29) | - | 0.88 (0.22–3.56) | - |

Notes:

Table A: a) Because there was no case for subfebrility, and diarrhoea, a COVID-19+ case was imputed for a male member of staff.

Significant odds ratios are displayed in bold ($\alpha \leq 0.05$).

Table B: a) Because there was no case for subfebrility, muscle ache, headache, diarrhoea, loss of smell/taste, and nausea, a COVID-19+ case was imputed.

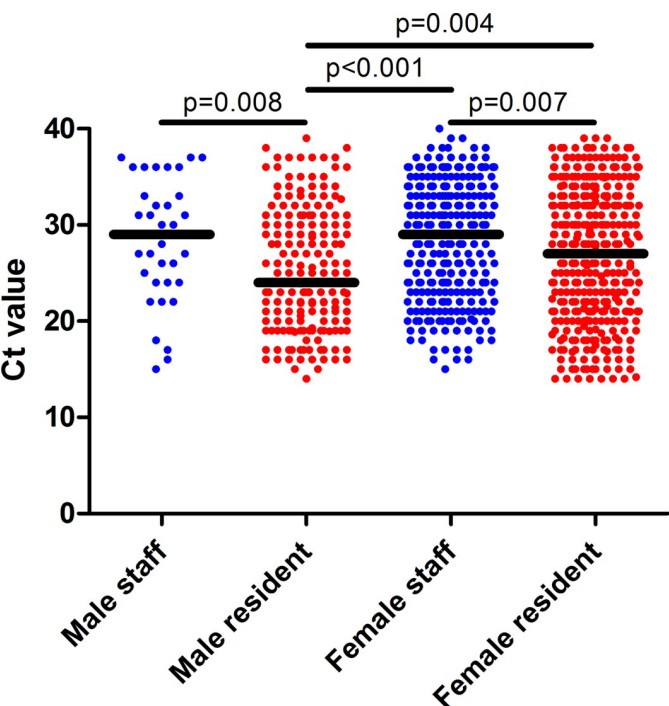

**Fig 3. The cycle threshold values (Ct-values) of symptomatic male staff (n = 34) and male residents (n = 189), as well as female staff (n = 287) and female residents (n = 422), the Netherlands, 27 February –1 June 2020.** Each dot represents an individual case. The black line shows the median Ct-value for the respective group.

SD: 6.3) compared to female residents (26.88, SD: 6.7, $\alpha \leq 0.01$). There was no statistically significant difference between the mean Ct-value for male and female staff.

## Multinominal multivariable regression analyses

The relationship between symptoms and Ct-values is further examined for staff in Table 2A and for residents in Table 2B, in which symptoms are stratified by sex and viral load levels (indicated by Ct-value) for cases with known test results and symptoms. For female staff, loss of taste and/or smell appeared less frequently in the medium viral load group (aOR 0.49, CI 0.26–0.91) and the high viral load group (aOR 0.15, CI 0.05–0.46) compared to female staff with low viral load. Because of the very low sample size for male staff per Ct-level, no multinominal regression analysis was performed for male staff.

For female residents, subfebrility appeared more frequently in the high viral load group (aOR 2.80 CI 1.12–6.), with the medium load group being borderline non-significant (aOR 2.41 CI 0.96–6.03). Finally, female residents with a medium and high viral load had increased odds of mortality compared to female residents with low viral load, with an OR of 2.00 (CI 1.04–3.83) and an OR of 3.08 (CI 1.62–5.88) respectively. This trend was not observed within male residents with medium and high viral loads, with an OR of 1.43 (CI 0.58–3.54) and an OR of 1.73 (CI 0.71–4.23) respectively. There were no reported deaths for staff within our study's time period.

## Discussion

This large study, which included LTCF residents and staff during the first COVID-19 wave in a Dutch region, suggests that subfebrility and fatigue may also be symptoms suggestive of COVID-19 in LTCF residents. Associated symptoms for LTCF staff were fever, cough, and

**Table 2.** a: Symptoms for female LTCF staff between low, medium, and high viral loads (indicative by cycle threshold value). b: Symptoms for LTCF residents between low, medium, and high viral loads (indicative by cycle threshold value), stratified for sex.

**A**

Female LTCF staff with known test result and symptoms (n = 1086)

| | No viral load (confirmed COVID-19-) (n = 807)[a] | | | Low viral load (CT>31) (n = 102)[a] | | | Medium viral load (22< Ct≤31) (n = 119)[a] | | | High viral load (CT≤22) (n = 58)[a] | | |
|---|---|---|---|---|---|---|---|---|---|---|---|---|
| | n/N[b] | % | OR (95% CI) | n/N[b] | % | OR (95% CI) | n/N[b] | % | OR (95% CI) | n/N[b] | % | OR (95% CI) |
| **Age in years (mean, standard deviation)** | 41.8, 13.7 | | 1.01 (0.96–1.06) | 44.0, 13.3 | | Ref. | 45.0, 13.2 | | 1.01 (0.99–1.03) | 44.7, 13.3 | | 1.01 (0.98–1.03) |
| **Symptoms** | | | | | | | | | | | | |
| Temperature increase | 321/807 | 39.8 | | 59/102 | 57.3 | Ref. | 71/119 | 59.6 | | 41/58 | 70.7 | |
| No increase | 486/807 | 60.2 | Ref. | 44/102 | 42.2 | Ref. | 48/119 | 40.3 | Ref. | 17/58 | 29.3 | Ref. |
| Subfebrility | 21/807 | 2.6 | 1.72 (0.21–12.95) | 1/102 | 1.0 | Ref. | 3/119 | 2.5 | 2.60 (0.26–26.40) | 1/58 | 1.7 | 2.24 (0.13–38.93) |
| Fever | 300/807 | 37.2 | **0.38 (0.25–0.60)** | 58/102 | 56.9 | Ref. | 68/119 | 57.1 | 0.94 (0.54–1.64) | 40/58 | 69.0 | 1.46 (0.72–2.96) |
| Cough | 545/807 | 67.5 | 0.64 (0.39–1.06) | 75/102 | 73.5 | Ref. | 91/119 | 76.5 | 1.07 (0.57–2.00) | 46/58 | 79.3 | 1.09 (0.49–2.42) |
| Muscle ache | 54/807 | 6.7 | **0.40 (0.21–0.76)** | 16/102 | 15.7 | Ref. | 21/119 | 17.6 | 1.14 (0.55–2.34) | 5/58 | 8.6 | 0.48 (0.16–1.40) |
| Loss of smell/taste | 51/807 | 6.3 | **0.10 (0.06–0.17)** | 35/102 | 34.3 | Ref. | 24/119 | 20.2 | **0.49 (0.26–0.91)** | 4/58 | 6.9 | **0.15 (0.05–0.46)** |

**B**

LTCF residents with known test result and symptoms (n = 698)

| | No viral load (confirmed COVID-19-) (n = 218) | | | | Low viral load (Ct > 31) (n = 130) | | | | Medium viral load (22<Ct≤31) (n = 188) | | | | High viral load (Ct≤22) (n = 162) | | | |
|---|---|---|---|---|---|---|---|---|---|---|---|---|---|---|---|---|
| | Male (n = 55) | | Female (n = 163) | | Male (n = 25) | | Female (n = 105) | | Male (n = 68) | | Female (n = 120) | | Male (n = 63) | | Female (n = 99) | |
| | n/N | OR (95% CI) | n/N | OR (95% CI) | n/N | OR (95% CI) | n/N | OR (95% CI) | n/N | OR (95% CI) | n/N | OR (95% CI) | n/N | OR (95% CI) | n/N | OR (95% CI) |
| **Age in years (mean, standard deviation)** | 81.5, 11.8 | 1.01 (0.96–1.06) | 85.7, 8.5 | 1.00 (0.97–1.03) | 81.0, 7.3 | Ref. | 85.8, 8.8 | Ref. | 83.0, 7.5 | 1.03 (0.98–1.09) | 83.9, 10.2 | 0.98 (0.95–1.01) | 82.2, 8.1 | 1.02 (0.97–1.07) | 85.0, 8.3 | 0.99 (0.96–1.02) |
| **Symptoms** | | | | | | | | | | | | | | | | |

*(Continued)*

**Table 2.** (Continued)

| | n/N | % | OR (95% CI) | n/N | % | OR (95% CI) | n/N | % | OR (95% CI) | n/N | % | OR (95% CI) | n/N | % | OR (95% CI) | n/N | % | OR (95% CI) | n/N | % | OR (95% CI) | n/N | % | OR (95% CI) |
|---|---|---|---|---|---|---|---|---|---|---|---|---|---|---|---|---|---|---|---|---|---|---|---|---|---|---|
| **Temperature increase** | 34/55 | 61.8 | | 158/163 | 96.9 | | 22/25 | 88.0 | | 94/105 | 89.5 | | 60/68 | 88.2 | | 101/120 | 84.2 | | 58/63 | 92.1 | | 78/99 | 78.8 | |
| No increase | 21/55 | 38.2 | Ref. | 44/163 | 27.0 | Ref. | 7/25 | 28.0 | Ref. | 32/105 | 30.5 | Ref. | 12/68 | 17.6 | Ref. | 23/120 | 19.2 | Ref. | 14/63 | 22.2 | Ref. | 22/99 | 22.2 | Ref. |
| Subfebrility [a] | 0/55 | 0.0 | Ref. | 5/163 | 3.1 | **0.08 (0.01–0.89)** | 3/25 | 12.0 | 0.33 (0.10–1.04) | 11/105 | 10.5 | Ref. | 8/68 | 11.8 | 1.18 (0.22–6.30) | 19/120 | 15.8 | 2.41 (0.96–6.03) | 5/63 | 7.9 | 0.66 (0.12–3.74) | 21/99 | 21.2 | **2.80 (1.12–6.96)** |
| **Fever** | 34/55 | 61.8 | Ref. | 114/163 | 69.9 | 0.53 (0.17–1.64) | 15/25 | 60.0 | 1.25 (0.72–2.19) | 62/105 | 59.0 | Ref. | 48/68 | 70.6 | 1.51 (0.47–4.82) | 78/120 | 65.0 | 1.75 (0.92–3.31) | 44/63 | 69.8 | 1.20 (0.38–3.76) | 56/99 | 56.6 | 1.39 (0.72–2.70) |
| **Fatigue** | 1/55 | 1.8 | Ref. | 4/163 | 2.5 | **0.06 (0.01–0.54)** | 5/25 | 20.0 | **0.21 (0.07–0.67)** | 12/105 | 11.4 | Ref. | 9/68 | 13.2 | 0.58 (0.17–1.99) | 14/120 | 11.7 | 1.10 (0.48–2.55) | 10/63 | 15.9 | 0.70 (0.21–2.38) | 17/99 | 17.2 | 1.61 (0.71–3.62) |

Notes:

Significant odds ratios are displayed in bold ($\alpha \leq 0.05$).

a) Because there was no case for subfebrility, a no viral load case was imputed for a male resident.

loss of tase and smell, and muscle ache, which are symptoms typical of COVID-19 [14,15,21]. Male residents seem to have an overall increased risk of mortality compared to female residents. Finally, female residents with medium to high viral load (Ct-values 31 or less) have an increased odds of mortality compared to female residents reporting low viral load.

Current literature on symptoms in LTCF residents and staff is plentiful. Several articles explicitly mention the variety and aspecificity of symptoms in LTCF residents [16,17,19,23]. There is an overlap in symptoms between COVID-19 negative and COVID-19 positive individuals [29], with fever, cough, hypoxia, dyspnea, and delirium being the most reported symptoms in residents with confirmed COVID-19 [17]. Although fever, cough, and dyspnea have been reported frequently in our study, our findings suggest that aspecific symptoms are more discriminative for a COVID-19 diagnosis than symptoms which are normally attributed to COVID-19 in LTCF residents, such as a loss of taste and/or smell, fever and cough. An explanation for the limited, although not unimportant, discriminative value in residents of these 'typical' COVID-19 symptoms, may be that these symptoms are caused by co-morbidities and increased frailty, which are typical for this population [4,9,38–41].

Our results show an increase in subfebrility symptoms for female residents as viral load increases, but not for fatigue. For staff, loss of taste and/or smell was more prevalent in the low viral load group compared to staff with medium and high viral load. Therefore, a loss of smell and/or taste might be a symptom which appears later on in the disease period. This has also been suggested by a systematic review on the onset and duration of loss of smell/taste in patients with COVID-19 [42]. Additionally, we saw an increase in CFR with an increase in viral load for female residents only. Several COVID-19 studies on viral load indicate that a higher viral load is associated with worse outcomes in COVID-19 patients, as well as an increased probability of severe disease prognosis and mortality [30,32,43].

To our knowledge, few studies have examined symptoms with regard to different levels of viral load [32,44]. A retrospective cross-sectional study established that high viral load was associated with more signs and symptoms at diagnosis and a more frequent pattern of respiratory and systemic symptoms among symptomatic hospitalized and outpatient COVID-19 cases. Cough and dyspnoea were also reported more often in cases reporting high viral loads compared to cases with lower viral loads [32]. These results differ from our findings, in which dyspnoea was not associated with COVID-19. The prevalence of cough in our study was the same among different Ct-levels for male and female staff. Another study found no correlation in CT-values between symptomatic and asymptomatic residents or staff, indicating that symptomatic and asymptomatic residents and staff had similar viral loads when infected with SARS-CoV-2 [44]. This might suggest that our findings may also be applicable for Ct-values found within asymptomatic residents and staff.

Due to the testing policy at the time, not all residents and staff were eligible for testing. However, out of the 696 tested residents who tested positive, 27% passed away relatively soon after their positive test. When we included the strongly suspected COVID-19 deaths reported by the LTCF physician, the CFR increased to 40%. By including these cases, we could estimate the total number of deaths among LTCF residents. However, this could have led to either an underestimation of the CFR due to a missing number of deaths with non-specific COVID-19 symptoms, or an overestimation due to a missing number of asymptomatic and/or not tested symptomatic cases. In the literature, CFRs for LTCFs within the first wave of the pandemic vary. CFRs ranged from 10% up to 35%, with an aggregated CFR of 23% [17]. This suggests that our CFR is probably an overestimation of the CFR.

On average, viral load was higher for residents than staff, in which male residents had a higher overall load compared to female residents. It is likely that the number of included male staff (n = 34) was insufficient to find a difference in viral load between male and female staff.

Additionally, our results show that male residents have an overall increased risk of mortality compared to female residents, which is in line with other studies, in which worse outcomes have been reported in men [45–48]. This difference between men and women was observed in studies on the SARS-CoV virus as well, in which men experienced higher CFRs compared to women [49,50]. This difference in viral load and mortality between men and women may be explained by several factors, including the host's genes, the host's microbiome, and the role of male and female sex hormones (SexHs) [45,51–54]. Estrogens, for example, activate both the innate and the adaptive immune responses and are therefore considered immuno-stimulatory, whereas testosterone is immuno-suppressive. As a result, women are able to clear pathogens more efficiently than men [55]. Because the production of SexHs differs in multiple stages of life, such as during pregnancy, menopause, and andropause, the difference in levels of SexHs may influence the immune response towards COVID-19. Therefore, SexHs influence the immune system response age-specifically [51]. With regards to the difference in viral load level between men and women, the literature seems unclear. Several studies reported varying viral load levels when comparing men and women; some reported women had higher load levels and some reported lower load levels compared to men, as well as studies reporting no difference in viral load level [56–58]. However, there seems to be support for an overall lower viral load level and COVID-19 outcomes for women, as the immune regulatory genes from the X chromosomes in women appear to cause lower viral load levels, inflammation and death after COVID-19 infections [57,59]. The differences in SexHs may have led to the found disparity between male and female residents. Other demographic factors such as age have also been linked to worse outcomes for COVID-19 and disproportionate infection rates. In our findings, age was significant only when comparing COVID-19 negative and COVID-19 positive staff, but not between different viral load levels within female staff, as well as male and female residents.

This study gave insights into COVID-19 testing within the Dutch LTCF setting during the first wave of the pandemic, by compiling a dataset of tested residents and staff from LTCFs in South Limburg. By creating a link between epidemiological data and microbiological data, we were able to further examine CFR and viral load. Because all test results were determined by a single laboratory, no inter-lab variance could have taken place, which increases the comparability of the Ct-values. Additionally, by using the same technique and diagnostic device for all samples, intra-lab variance was limited.

The analyses of our data did come with some limitations. Due to the symptom-based testing policy at that time, we did not test for asymptomatic patients. During the pandemic, it became clear that asymptomatic and presymptomatic cases can have a significant role in transmitting SARS-CoV-2. Hence, our epicurve will be incomplete, and our CFR biased. By requesting LTCFs to make an inventory of suspected COVID-19 deaths, we were able to estimate a portion of non-lab-confirmed COVID-19 related deaths. Additionally, because these data were collected in a period of (inter)national crisis, data collection was not as streamlined compared to planned studies. As a result, we were not able to link cases with the LTCF in our dataset. Therefore, we could not correct for any LTCF specific testing policies later on in the pandemic. This also means that we could not compare the number of included residents to the total number of residents (beds) and staff. Furthermore, because little was known of SARS-CoV-2, guidelines and measures were adapted as new information became available. This resulted in the suspected case definition for COVID-19 to change during the study period. Because testing policy changed during our study's time period, the composition of included staff and residents may have influenced the ORs of the included symptoms. To estimate the effect testing policy may have had on the composition of included residents and staff, we conducted a multivariable model with an added variable for testing policy periods (27 Feb—5 April, 6 April-16 April, 17 April-1 June). The interactions between testing policy period and symptoms were

limited and did not influence our found effects. Nevertheless, symptoms besides fever, dyspnoea, and cough, were not included from the start, as they were not included in the initial suspected case definition. One should therefore take into account the overrepresentation of symptoms such as fever, dyspnoea, and cough. Finally, because only one specimen was collected for the majority of cases, the interpretation of the Ct-value was challenging as it could represent the start, the middle, or the end of the disease period. Nevertheless, most test were done as closely to the onset of symptoms–within the first days–so reasonable compatibility is expected. One should note that the results discussed in this study are on the wildtype Wuhan strain of SARS-CoV-2. Ct-values seem to differ per virus variant; higher viral loads have been observed in the alpha, beta, and delta variants of SARS-CoV-2 compared to the wildtype variant [60,61], although significantly lower than previously reported [62]. It is likely that (future) different variants will have different levels of viral load compared to the wildtype variant examined in this study. However, the relationship between the level of viral load and the appearance of symptoms may still be applicable. Future studies are needed to verify this. Furthermore, the effect of vaccination and previous infection may influence the viral load, as well as which symptoms a person may have in future infections.

Future studies could further improve our understanding of Ct-values and viral load by performing multiple measurements of the Ct-value throughout the disease period, which would clarify the change in viral load at different points in the diseases period.

## Conclusions

Symptoms associated with COVID-19 seemed to differ between LTCF staff and residents. Although current suspected case definitions are sufficient for LTCF staff, aspecific symptoms including subfebrility and fatigue seem to be associated with COVID-19 in LTCF residents. Even though aspecific symptoms are not typical for COVID-19, these aspecific symptoms should be taken into account when checking up on residents and physicians should consider testing residents for COVID-19 when these symptoms do occur.

Male residents reported higher viral loads compared to female residents and had an overall increased risk in mortality, whereas female residents only had an increase in mortality per viral load level. Men, as well as female COVID-19 positive residents with medium to high viral load (Ct-values 31 or less) should be monitored closely, as these residents have an increased odds of mortality.

## Supporting information

**S1 Table. Descriptive statistics of staff and resident characteristics by test result.** a) Sex was unknown for 6 residents. b) Age was unknown for 2 staff and 4 residents. c) $\alpha \leq 0.01$. d) Comorbidities were unknown for 554 COVID-19- residents, and 24 COVID-19+ residents, for a total of 578 residents. e) $\alpha \leq 0.001$.
(DOCX)

**S2 Table. Descriptive statistics of staff (n = 4) and resident (n = 163) characteristics with unknown COVID-19 test result.**
(DOCX)

## Acknowledgments

**Declarations**

The authors would like to thank all healthcare staff of the participating LTCFs and the Public Health Service South-Limburg for the data collection, along with all Maastricht UMC+

laboratory staff involved in performing the COVID-19 testing. We would also like to thank Brian van der Veer from Maastricht UMC+ for his assistance in gathering the Ct-values.

## Author Contributions

**Conceptualization:** Mitch van Hensbergen, Casper D. J. den Heijer, Christian J. P. A. Hoebe.

**Data curation:** Mitch van Hensbergen.

**Formal analysis:** Mitch van Hensbergen, Suhreta Mujakovic, Nicole H. T. M. Dukers-Muijrers, Petra F. G. Wolffs, Inge H. M. van Loo.

**Methodology:** Mitch van Hensbergen, Casper D. J. den Heijer, Christian J. P. A. Hoebe.

**Supervision:** Casper D. J. den Heijer, Nicole H. T. M. Dukers-Muijrers, Christian J. P. A. Hoebe.

**Visualization:** Mitch van Hensbergen.

**Writing – original draft:** Mitch van Hensbergen.

**Writing – review & editing:** Mitch van Hensbergen, Casper D. J. den Heijer, Suhreta Mujakovic, Nicole H. T. M. Dukers-Muijrers, Petra F. G. Wolffs, Inge H. M. van Loo, Christian J. P. A. Hoebe.

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
