## [Decision Letter · Decision Letter 0]

11 Apr 2022

PONE-D-22-04854Evaluation of symptomatology and viral load among residents and healthcare staff in long-term care facilities: A coronavirus-19 cohort studyPLOS ONE

Dear Dr. van Hensbergen,

Thank you for submitting your manuscript to PLOS ONE. After careful consideration, we feel that it has merit but does not fully meet PLOS ONE’s publication criteria as it currently stands. Therefore, we invite you to submit a revised version of the manuscript that addresses the points raised during the review process.

Both reviewers agree that your manuscript requires major changes. Please address all of their constructive suggestion for changes before resubmitting

We look forward to receiving your revised manuscript.

Kind regards,

Joël Mossong

Academic Editor

PLOS ONE

Journal Requirements:

2. During your revisions, please confirm whether the wording in the title is correct and update it in the manuscript file and online submission information if needed. Specifically, we believe that "coronavirus-19" should be corrected to either "COVID-19" or "coronavirus disease 2019".

4. Ethics statement only appears at the end of the manuscript:

Your ethics statement should only appear in the Methods section of your manuscript. If your ethics statement is written in any section besides the Methods, please move it to the Methods section and delete it from any other section. Please ensure that your ethics statement is included in your manuscript, as the ethics statement entered into the online submission form will not be published alongside your manuscript. 

Reviewers' comments:

Reviewer's Responses to Questions

**Comments to the Author**

1. Is the manuscript technically sound, and do the data support the conclusions?

Reviewer #1: Partly

Reviewer #2: Partly

2. Has the statistical analysis been performed appropriately and rigorously? 

Reviewer #1: No

Reviewer #2: No

3. Have the authors made all data underlying the findings in their manuscript fully available?

Reviewer #1: No

Reviewer #2: Yes

4. Is the manuscript presented in an intelligible fashion and written in standard English?

Reviewer #1: No

Reviewer #2: No

5. Review Comments to the Author

Reviewer #1: This study assessed differences in symptoms between test-negative and test-positive symptomatic LTCF staff and residents during the first wave of the COVID-19 pandemic. It also assessed the relationship between Ct values and variables symptoms, gender and outcome.

While the descriptive results of this large case series are certainly valuable, there are some issues with the analytical results that would need to be addressed.

1) The distribution of symptoms in the comparison groups (test-negative staff and residents) is likely to depend on the testing strategy, therefore the association of symptoms with test-positivity may also depend on the testing strategy. In order to investigate this potential effect, the authors should consider 1) adding a variable with the three different testing policy periods (27 Feb-5 April/6 April-16 April/17 April-1 June) in the multiple logistic regression model (and also test for interaction with this variable), and; 2) as testing policies are not necessarily applied in the same way in each LTCF, add an LTCF variable, either as fixed effect or as random effect in a mixed effects model.

2) The completeness of data on symptoms is very different between staff and residents and between test-negatives and test-positives. While for staff, symptoms were unknown for only 48 (5.1%) covid-19 negatives and 8 (1.6%) covid-19 positives, in residents the number of unknowns was much higher with 623 (73.8%) and 141 (20.3%), respectively. This differential distribution of missing values may of course also affect the observed associations. In the discussion the authors say “Due to the symptom-based testing policy at that time, we did not test for asymptomatic patients.” , so missing data on symptoms would not be due to testing of asymptomatic residents (in e.g. “vulnerable groups”)? On L89, the authors report for the period since 17 April “Although negatively tested cases were also reported to the PHS by the LTCFs, no additional information on comorbidities or symptoms was provided by the LTCFs.” Doesn’t this mean that the analysis of symptoms should be restricted to the period before 17 April?

3) It is unclear why most analyses were stratified by sex. Was this because a statistically significant interaction with sex was observed when pooling the data? If so, this should be mentioned. One the one hand, stratifying reduces statistical power (especially for male staff and residents) and on the other hand multiplying the number of analysed associations by two increases the likelihood of finding statistically significant results by chance. In several cases, the difference in aORs between sexes seems to be not significant. Is the stratification by gender always justified? Another point, the lower Ct values found in male compared to female patients is an observation that is not very consistent in literature and this should be better addressed in the discussion. The higher mortality in males is quite consistent though (and not only for COVID-19).

Furthermore, please have the manuscript reviewed by a native speaker as there is some room for improvement of the English language.

Specific comments:

Abstract

See comments below

Specifically:

L37. Therefore, physicians should consider testing residents even when only aspecific symptoms are present.

Actually, the ECDC testing guidance recommended to test all residents and staff (symptomatic or not) once there was/is one confirmed case of COVID-19 in the LTCF.

Introduction

L 48. the proportion of COVID-19-related deaths still ranges between 0% and 62% in LTCF residents in European countries, as reported up until November 9th 2021 [8].

Reference 8 was updated on 21 Feb 2022 and doesn’t show these data anymore.

L 51. LTCF residents often have chronic diseases, are mentally incapable and have complex health needs

Suggest “LTCF residents often have chronic diseases, mental impairment and complex health needs”

(the “often” should apply to all three conditions)

L71. During the first wave of the COVID-19 pandemic, almost all (suspect) COVID-19 cases for residents and staff were tested by the Public Health Services (PHS) in the South Limburg region. Additionally, all LTCF tests not performed by our PHS were mandatory notified. With this virtually fully complete LTCF dataset of the classic SARS-CoV-2 variant within one region, we evaluated which symptoms are associated with COVID-19 in LTCF staff and residents, as well as assessed the relationship of viral load with symptoms and case fatality rate (CFR).

Most of this is repeated afterwards (e.g. L85 and following. L 105). Suggest to replace with “The objectives of our study were to assess symptoms associated with COVID-19 in LTCF staff and residents and to assess the relationship between viral load and COVID-19-related symptoms and case fatality, in LTCF residents and staff in the South Limburg region.”

L73. Additionally, all LTCF tests not performed by our PHS were mandatory notified.

PCR tests or rapid antigen tests ? If RADTs, were they confirmed by PCR?

“mandatorily” rather than mandatory

L77. Our assessment of the first COVID-19 wave in LTCFs in the Netherlands has contributed to the limited research on Ct-values as an indicator of viral load and it’s relation with COVID-19 symptomatology and mortality.

Delete sentence (or move to conclusion)

L 82. We performed a cross-sectional epidemiological and laboratory analysis of LTCF residents and staff who were tested for COVID-19 in South Limburg, the Netherlands, from February 27th 2020 to June 1st 2020.

Not cross-sectional, rephrase. The title mentions cohort study, but it’s not a real cohort study either as non-tested staff and residents are not included. Title should be adapted as well e.g. “COVID-19 symptomatology and viral load among residents and healthcare staff in long-term care facilities in

How many LTCFs were included?

L96. In response to the increase of COVID-19 cases within LTCFs, additional infection control measures were implemented by the LTCFs, among which a (temporarily) halt of admission of new residents and a limited number of visitors, or total visitor ban. LTCF staff wore gowns, surgical mouth masks, and gloves when they expected to get within 1.5 meters of COVID-19 positive residents or their surroundings.

… (temporary) admission stop , limitation of the number of visitors or a total visitor ban, depending on the LTCF (?)

surgical mouth masks -> medical face masks

L 108. Additionally, a suspected case should have a history of travel or residence in a country/area reporting local or community transmission, or have had to be in close contact with a confirmed, or probable COVID-19 case in the past two weeks [33, 34].

Was South Limburg was considered an area reporting community transmission at the time? If so, from when?

L111. As of April 6th 2020, testing policy was expanded. Whereas testing was first reserved to cases reporting serious health symptoms, tests were now also performed within vulnerable groups when this was necessary to be able to provide care or treatment. Tests were also available for healthcare staff when they showed symptoms indicative of COVID-19, and when they were essential in providing care. As of April 17th, LTCFs had the option of performing COVID-19 tests on residents themselves.

The change of testing policy on 6 April should be mentioned in the section “Study design, setting and test policy”

What do you mean with “vulnerable groups” in the context of an LTCF? Aren’t all residents vulnerable because of their age? So there was never a policy of testing all residents from February 27th 2020 to June 1st 2020?

The case definition as such is actually not clearly mentioned in this section

L154. However, the date of onset for the epicurve was not available for every case. Therefore, we chose the date which resembled the date of onset closest. We prioritized the dates as followed: day of onset, day of testing, day of test result, day of communication to the PHS.

Results

L193. When we included the highly suspected COVID-19 deaths, the overall CFR increased to 40.4% (347/859), of which 53.6% (159/297) among men and 33.5% (188/561) among women.

What is the denominator here exactly (n=859)?

In addition, it would be important to know the total population of residents and staff in the included LTCFs for the study period (tested and non-tested)? Alternatively for residents, the number of beds in the included LTCFs. Also the number of LTCFs seems to be missing.

L200. Multivariate -> multiple or multivariable

L209-211:

-What does the Note a (“Unless stated otherwise”) mean?

-Note c in Table 1a and note d in Table 1b on the "creation of an artificial case" is unclear

L237-L242: suggest to remove the line Symptoms “yes” from the Tables, that’s implicit

Discussion

See general comments, which may have an impact on the discussion.

The study limitations section should be further developed

L342. By requesting LTCFs to make an inventory of deaths who were suspect of COVID-19, we were able to estimate a portion of asymptomatic COVID-19 related deaths.

How can suspected COVID-19-related deaths (I assume not lab-confirmed but with a clinical picture compatible with COVID-19) be asymptomatic?

L353. However, the variance in Ct-value between variants is limited, therefore upholding the findings of this study [56].

Evidence regarding variant-dependent viral load is still evolving (e.g. Omicron), so this statement is too strong.

Also the influence of vaccination and previous infection should be mentioned, not only regarding Ct values/viral load but also regarding symptoms.

Reviewer #2: Major comments

This manuscript describes symptoms and viral load among residents and staff members of long-term care facilities in the regional public health service Limburg Zuid in the Netherlands between February 2020 - June 2020. The combination of symptoms and viral load is scarce or not available in most countries and this study therefore contributes to the scientific literature.

The authors stratify the results by sex throughout the manuscript and this is somewhat an odd choice since no significant differences between males and females were observed (all 95%CIs overlap). The number of significantly COVID-19 associated symptoms differs by sex, but this seems to be a sample number issue due to the lower number of male residents and male staff members. The authors mention in the results a significant difference in mortality between males and females, but do not show these results in the presented tables.

A major concern is that the case definition changed over the course of the study and that the study outcomes (i.e. fever and respiratory symptoms) were part of the case definition (at least in the first phase). This makes the presented ORs between the cases and controls difficult to interpret. I.e. cases with subfebrility could not be found in the first phase since those were excluded based on the case definition. The authors do not discuss this limitation in the discussion.

The authors used the forced entry model selection method for the multivariable logistic regression model and included all variables from the univariate analysis with a p-value above 0.1 to the model. Although multiple variable selection methods are used in the field, this does not seem to be the best choice. Multiple variables were added to the model which do not significantly contribute to the model (based on the provided information), and make the model unnecessary complex. It might be better to use a mixture of forward and reverse variable selection together with AIC or BIC. Also: two different models were used for males and females (i.e. the model in table 1 shows age + 3 variables for males and age + 5 variables for females), raising questions whether both models can be compared.

Minor comments:

Line 3: use official nomenclature throughout the manuscript: “COVID-19” for the disease and “SARS-CoV-2” for the virus.

Line 3: the term cohort study is misused here in my opinion.

Line 23: write “LTCF” out in full when first used in the abstract

Line 46: I suggest to use “wildtype SARS-CoV-2 Wuhan strain” instead of “classic SARS-CoV-2 virus”.

Line 50: “up” can be omitted. I suggest to add the reporting organisation (European Centre for Disease Prevention and Control) to the sentence.

Line 52-59: these sentences need to be rephrased and can be shortened since they convey the same message twice. Also: rapid antigen assays and availability of RT-PCR testing have made it easier to recognize COVID-19 outbreaks in LTCFs.

Line 86: please describe what “information about the person to be tested” was collected by phone.

Line 112: what is meant by serious health symptoms?

Line 113: be specific. In which situations was it necessary to test vulnerable groups to be able to provide care or treatment? Did the LTCFs still use a case definition in this period?

Line 114: which symptoms? Fever still included in the case definition?

Line 115-116: Did the LTCFs use a case definition in this period? What about screening of contacts of cases? I am asking since you barely have asymptomatic cases in the dataset.

Line 169: I think the authors mean “forced entry method” instead of “forced enter method”

Line 259-260: In my view aspecific symptoms are by definition not discriminative.

Line 302: “original” can be omitted

Line 317 – 325: I suggest to omit “Although race…ethnic makeups” from the manuscript. This topic was not part of the study and statements on human race should be avoided in scientific publications.

Line 326: “complete picture” sounds like an unusual statement for a scientist.

Line 326-337: this paragraph reads like it was written by the sales manager of the PHS.

Line 339: a few individuals without symptoms are in the dataset though?

Line 349-350: which role could Ct values have in the clinical decision making as reported by the referenced papers? And what does this study confirm or add to the existing literature?

Line 351-354: Contradictio in terminis: “Knowing that the Ct-value differs per virus variant” and “variance in Ct-value between variants is limited”. Does it differ or not? Or not yet a scientific consensus?

Line 534 ref 33: this is only a link to the general RIVM web archive website. Please add the information to the specific document.

Figure 2: Please add figure legend or describe the meaning of the colours in the figure description.

Tables: please describe or indicate the meaning of bold text.

Table 1B: Why is the unadjusted OR for female and fever not in bold? P-value above 0.1? Why added to the multivariable model?

Table 2a: Please add the notes in Table 2A. They cannot be similar to those in Table 2B.

6. PLOS authors have the option to publish the peer review history of their article (what does this mean?). If published, this will include your full peer review and any attached files.

Reviewer #1: No

Reviewer #2: No

---

## [Author Response · Author response to Decision Letter 0]

25 May 2022

We would like to refer to our document titled 'Response to Reviewers' for our response to all editor and reviewer comments.

---

## [Decision Letter · Decision Letter 1]

26 Jul 2022

PONE-D-22-04854R1Evaluation of symptomatology and viral load among residents and healthcare staff in long-term care facilities: a coronavirus disease 2019 retrospective case-cohort studyPLOS ONE

Dear Dr. van Hensbergen,

Thank you for submitting your manuscript to PLOS ONE. After careful consideration, we feel that it has merit but does not fully meet PLOS ONE’s publication criteria as it currently stands. Therefore, we invite you to submit a revised version of the manuscript that addresses the points raised during the review process.

ACADEMIC EDITOR:While the manuscript has improved there remain a number of issues that need to be addressed and clarified. Please address all of them before resubmitting.==============================

We look forward to receiving your revised manuscript.

Kind regards,

Joël Mossong, PhD

Academic Editor

PLOS ONE

Reviewers' comments:

Reviewer's Responses to Questions

**Comments to the Author**

1. If the authors have adequately addressed your comments raised in a previous round of review and you feel that this manuscript is now acceptable for publication, you may indicate that here to bypass the “Comments to the Author” section, enter your conflict of interest statement in the “Confidential to Editor” section, and submit your "Accept" recommendation.

Reviewer #2: All comments have been addressed

2. Is the manuscript technically sound, and do the data support the conclusions?

Reviewer #2: Partly

3. Has the statistical analysis been performed appropriately and rigorously? 

Reviewer #2: Yes

4. Have the authors made all data underlying the findings in their manuscript fully available?

Reviewer #2: Yes

5. Is the manuscript presented in an intelligible fashion and written in standard English?

Reviewer #2: No

6. Review Comments to the Author

Reviewer #2: General:

Please omit “In doing so,” from the manuscript (3x)

Although significantly improved since the last submitted version, the description of the case definitions and testing policies can still be improved. I suggest to add one paragraph on the case definitions and testing guidelines and merge the respective sections from the paragraphs “Study design, setting and testing policy” and “Suspect case definition”. Please describe clearly how these different policies were applicable to LTCF residents (it seems like there were some overall National guidelines + some additional specific LTCF policies). Please add the policies for screening of LTCF residents who have had contact with a confirmed SARS-CoV-2 case (if those were included in the analysis) as well.

It is sometimes challenging to understand what the authors mean since the English is sometimes weak or is not always written in scientific language. I feel it is not my job to correct for this and I strongly suggest to seek support if this knowledge is not available among the authors. I made some comments and suggestions to improve (see minor comments below), but would like to stress here that the manuscript requires an additional thorough editing to make it readable.

The methods are a bit messy with sometimes very long paragraphs. Statistical analysis is almost 2 pages long for example and contains details that may not be very important for the reader (i.e. how variables are named or coded).

Minor comments

Line 74: replace ‘assess’ by ‘describe’

Line 77 – 79: this sentence is redundant and can be omitted

Line 83: replace “COVID-19” by “SARS-CoV-2”

Line 85: “Testing” = “SARS-CoV-2 testing”

Line 85: “reporting serious health symptoms, meaning whenever a case was sick enough to be” can be omitted.

Line 85: Were all LTCF residents admitted to the hospital tested or was this indicated based on symptoms as well? Please clarify.

Line 85-86: use ‘patient’, ‘individual’, or ‘LTCF resident’ in your case definitions rather than ‘case’. You can use ‘case’ after you have described your case definition(s).

Line 87-91: Do you mean that all LTCF residents 70+ were regularly and routinely tested? Were symptoms included in this case definition? How frequently were the residents tested if symptoms were not included?

Line 91: omit ‘also’

Line 100: I assume these were SARS-CoV-2 antigen assays? Please specify and be precise. Sample collection by the LTCF or sample testing? LTCF can’t do PCR testing.

Line 115: “Suspect” = “Suspected”

Line 136-145: “After merging these data…in Figure 1)”: these are result

Line 181-182: Not clear what was done for 959 cases (30%) and what happened to the remaining 70%?

Line 190-195: I suggest to omit “For some symptoms…on other variables” and add a foot note to the respective table.

Line 196-202: move to discussion

Line 218: the first paragraph of the results section does not have a subheading, while other paragraphs have.

Line 222-223: write in past sense: “these data were not included in further analyses”

Line 223-225: you may would like to say something about Tables S1 and S2. Why do you show these tables, what should the reader catch from the data?

Line 226-230: please add yes/no statistically different between male and female (2 times)

Line 286: ‘relevant’ for what?

Line 288: Does that mean that non-associated symptoms are not typical for COVID-19 (i.e. runny nose)?

Line 288: try to refer to original peer-reviewed publications rather than websites.

Line 294-295: “between COVID-19 negatives and COVID-19 positives” = “between individuals tested SARS-CoV-2 negative and individuals tested SARS-CoV-2 positive”

Line 297: “but our findings suggest aspecific symptoms to be more suggestive for COVID-19 test in residents” : I don’t understand what the authors are trying to say here. Please rephrase.

Line 305-306: replace “,which is in line with findings from” by “as suggested by”

Line 365-366: A large sample size does not mean that conclusions can be generalized to LTCFs in other regions or countries. The population of LTCFs in other regions/countries might be very different from those in the South of Limburg.

Line 366-373: I suggest to omit these lines from the discussion since it is (or must be) clear from your methods that all specimens were tested by the same PCR platform and laboratory. I think you can say that it is likely that the CFR is somewhere between the estimated CFR without and estimated CFR with the suspected deaths, but I don’t think it is correct that adding suspected deaths to your analysis have improved the accuracy of the estimate.

Line 383: explain “prepared studies”

Line 393: “test was performed” = “specimen was collected”

Line 395: you could have included days since symptom onset on the x-axis of Figure 3.

Line 395-397: you need to show the data if you make statements like “most test were done in connection to the onset of symptoms” and “so reasonable compatibility might be expected”. Both phrases are not correct English so need to be rephrased.

Line 399: “the SARS-CoV-2 virus” = “SARS-CoV-2” (and change this throughout the manuscript). The “V” in SARS-CoV-2 means “virus”

Line 400: add reference for limited Ct-value differences among variants. Is this the scientific consensus? I thought studies had found significant differences in the viral load between SARS-CoV-2 variants.

Line 400-402: The authors refer here to their own study showing a 1000 fold higher viral load for Delta compared to non-VOC strains. How can you say that the variance in viral load between variants seem to be limited?! What do other studies find? And how can you make a prediction on this topic for the future?

Line 403: do symptoms have an effect on viral load or vice versa?

Line 403-404: Sorry, I don’t understand what the authors mean by “as symptoms, which a person may have in COVID-19 infections following the first COVID-19 wave.”

Figure 2: y-axis: “COVID-19” instead of “COVID”

7. PLOS authors have the option to publish the peer review history of their article (what does this mean?). If published, this will include your full peer review and any attached files.

Reviewer #2: No

---

## [Author Response · Author response to Decision Letter 1]

9 Sep 2022

We kindly refer to our document titled 'Response to Reviewers v2.docx' for our response to the reviewer.

---

## [Decision Letter · Decision Letter 2]

14 Oct 2022

Evaluation of symptomatology and viral load among residents and healthcare staff in long-term care facilities: a coronavirus disease 2019 retrospective case-cohort study

PONE-D-22-04854R2

Dear Dr. van Hensbergen,

We’re pleased to inform you that your manuscript has been judged scientifically suitable for publication and will be formally accepted for publication once it meets all outstanding technical requirements.

Kind regards,

Joël Mossong, PhD

Academic Editor

PLOS ONE

Additional Editor Comments (optional):

Reviewers' comments:

Reviewer's Responses to Questions

**Comments to the Author**

1. If the authors have adequately addressed your comments raised in a previous round of review and you feel that this manuscript is now acceptable for publication, you may indicate that here to bypass the “Comments to the Author” section, enter your conflict of interest statement in the “Confidential to Editor” section, and submit your "Accept" recommendation.

Reviewer #2: All comments have been addressed

2. Is the manuscript technically sound, and do the data support the conclusions?

Reviewer #2: Yes

3. Has the statistical analysis been performed appropriately and rigorously? 

Reviewer #2: Yes

4. Have the authors made all data underlying the findings in their manuscript fully available?

Reviewer #2: Yes

5. Is the manuscript presented in an intelligible fashion and written in standard English?

Reviewer #2: Yes

6. Review Comments to the Author

Reviewer #2: Only two small suggestions for a textual change:

Line 27: I think something is missing here. “all’ = “SARS-CoV-2 tested”?

Line 376: I suggest to replace “discussed” by “obtained” and replace “are on the wildtype Wuhan” by “are based on infections with the wildtype Wuhan”

7. PLOS authors have the option to publish the peer review history of their article (what does this mean?). If published, this will include your full peer review and any attached files.

Reviewer #2: No

---

## [Editor Report · Acceptance letter]

21 Oct 2022

PONE-D-22-04854R2 

Evaluation of symptomatology and viral load among residents and healthcare staff in long-term care facilities: a coronavirus disease 2019 retrospective case-cohort study 

Dear Dr. van Hensbergen:

I'm pleased to inform you that your manuscript has been deemed suitable for publication in PLOS ONE. Congratulations! Your manuscript is now with our production department. 

Kind regards, 

on behalf of

Dr. Joël Mossong 

Academic Editor

PLOS ONE